# Dehydration regulates structural reorganization of dynamic hydrogels

Dan Xu[1], Xintong Meng[1], Siyuan Liu[1], Jade Poisson [1], Philipp Vana[2] & Kai Zhang [1,3] ✉

The dehydration process is widely recognized as a significant phenomenon in nature. Hydrogels, which are important functional materials with high water content and crosslinked networks, encounter the issue of dehydration in their practical applications. Here, we report the distinctive anisotropic dehydration modality of dynamic hydrogels, which is fundamentally different from the more commonly observed isotropic dehydration of covalent hydrogels. Xerogels derived from dynamic hydrogel dehydration will fully cover a curved substrate surface and exhibit hollow structures with internal knots, in contrast to the bulk xerogels produced by covalent hydrogel dehydration. Depending on the competing cohesion of polymer chains and the adhesion at the hydrogel-substrate interface, the previously overlooked reorganization of polymer networks within dynamic hydrogels, triggered by dehydration-induced stress, has been discovered to regulate such macroscopic structural reconstruction for dynamic hydrogel dehydration. With the attached hydrogel-substrate interface, the surface microstructures of substrates can also be engraved onto xerogels with high resolution and on a large scale. This work will greatly enhance our understanding of the soft matter dehydration process and broaden the applications of dehydration technologies using water-containing materials.

Hydrogels, soft materials with various water contents, have attracted much attention due to their intrinsic viscoelastic properties, porous structures, and processability[1,2]. The majority of water present in hydrogels is typically found in a "free" or unbound state[3,4]. Therefore, the primary challenge of using hydrogels in practice is dealing with dehydration[5]. The variation of water content in hydrogels during dehydration causes the collapse of capillaries in the porous domains and changes the interactions between different components. For example, it affects the interaction between polymer chains and water, the status of water, and the macrostructure of the materials[6-8]. Isotropic shrinkage of hydrogels with permanent networks during dehydration is prevalent, which is in agreement with theoretical predictions[9,10]. Consequently, the dehydration-relevant stress and

relaxation change were simply ignored for the isotropic shrinkage of covalent hydrogel dehydration[11,12]. In contrast, based on the dehydration of corn kernels[13] or polymer coatings[14], it has been demonstrated that the dehydration-induced stress can profoundly influence the resulting structure and therefore accompanying functions of water-containing materials. However, dynamic hydrogels, distinguished by their reversible networks, possess physicochemical properties analogous to those of organisms and exhibit adaptive responses to stress[15,16]. Pitifully, the current understanding of hydrogel dehydration is primarily based on static systems. This has rendered it challenging to investigate macroscale transitions in dynamic hydrogels, which are more common in living systems and functional materials.

[1]Sustainable Materials and Chemistry, Department of Wood Technology and Wood-based Composites, University of Göttingen, Göttingen, Germany. [2]Institute of Physical Chemistry, University of Göttingen, Göttingen, Germany. [3]Biotechnology Center (Biotechnikum), University of Göttingen, Göttingen, Germany. ✉e-mail: kai.zhang@uni-goettingen.de

## Results

In previous studies, the link between molecular behaviors and the macroscopic mechanical properties of dynamic hydrogels has been demonstrated[17]. A scaling relationship between water content and the cohesion or adhesion energy for double network hydrogels that encompass covalent and dynamic crosslink sites has been observed[18]. In light of this, we intend to provide a precise description of a dehydration-induced gradient stress distribution for the dynamic hydrogel system, and more importantly, the unique phenomenon derived from such a dynamic system which has not previously been considered in covalent hydrogel dehydration. The utilization of dynamic hydrogels will facilitate a deeper comprehension of the highly dynamic processes of dehydration. Moreover, the presence of dehydration-induced stress in dynamic hydrogels will allow reorganization of the dynamic crosslink sites on the microscale which will result in the structural reconstruction on the macroscale.

Here, we investigate a general dehydration process of dynamic hydrogels of varying viscoelasticity. The dynamic network is constructed by boronate esters, as illustrated in Supplementary Fig. 1. Compared with covalent hydrogels, the typical rheological behaviors of dynamic hydrogels mainly differ in the low-frequency zone. Specifically, a crossover point of the storage modulus (G′) and loss modulus (G″) only emerged in dynamic hydrogels (Supplementary Fig. 2). The dehydration of these two types of hydrogels in tubes interestingly led to two different modalities of the dehydration process and distinct structural features within the resulting xerogels (Fig. 1a). The transition process can be characterized by the resulting geometric parameters, which are referred to as the isotropy index (I.I.) and described by the equations below:

$$\text{I.I.} = \frac{R_1/R_0}{L_1/L_0} \qquad (1)$$

or

$$\text{I.I.} = \frac{2D/R_0}{L_1/L_0} \qquad (2)$$

Here, Eq. 1 refers to the isotropic modality, while Eq. 2 refers to the anisotropic modality. For covalent hydrogels, an isotropic shrinkage behavior (I.I. in the order of $10^0$) is generally observed[11,19]. The covalent hydrogels and inner wall underwent gradual separation during dehydration, accompanied by isotropic shrinkage, ultimately resulting in bulk xerogels. In contrast, with a continuous migration of air-hydrogel interface, dynamic hydrogels maintained a fully covered interface along the inner wall throughout the dehydration process, generating the hollow xerogel tubes with knots (I.I. is in the order of $10^{-2}$). Interestingly, these hollow xerogels had similar densities of about $1.1\,\text{g/cm}^3$ for both walls and knots (Supplementary Fig. 3). Additionally, the formation of these distinct and heterogeneous materials occurred independently of the hydrophobicity of the substrates (Supplementary Fig. 4, 5). Here, such a unique macroscopic structural reconstruction during anisotropic dehydration for dynamic hydrogels reveals the critical role of the dynamic network when compared to the static network observed in covalent hydrogels.

The resulting anisotropic hollow xerogel tubes exhibit specific microstructures (Fig. 1b). In particular, the hollow xerogels possess evenly scattered nanoscale pores, resembling those found in lyophilized dynamic hydrogels (Supplementary Fig. 6, 7). The diameter of these pores was ~50-60 nm (Supplementary Fig. 8). The wall of the hollow xerogel tubes was 110-120 μm thick, while the knots can have a thickness of up to ~300 μm. Apart from the pores, the xerogels have a homogeneous internal structure with no other microscopically ordered structures observed in OM or SEM images (Fig. 1c). The homogeneous structure was further verified by polarized Raman

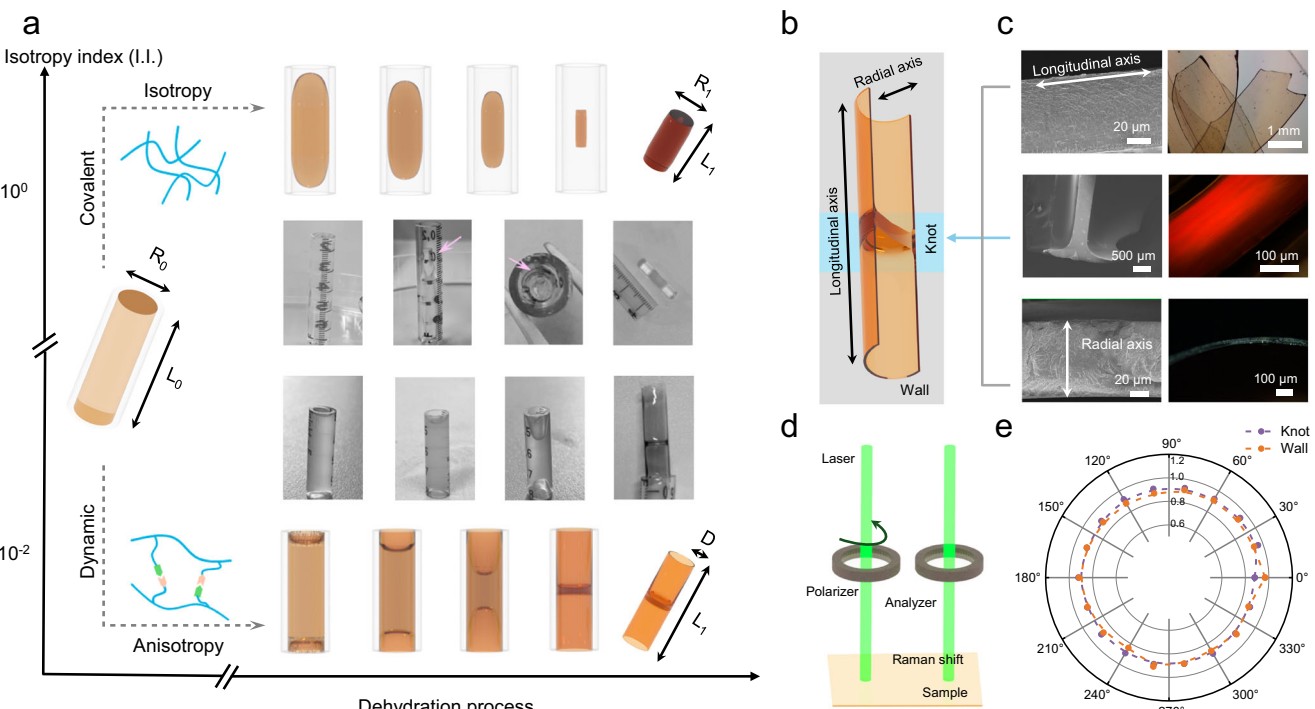

**Fig. 1 | Dynamic polymer network regulates the dehydration behaviors of hydrogels. a** Two typical dehydration behaviors of hydrogels. **b** Illustration of resulting hollow xerogel tubes from dynamic hydrogels. **c** The optical microscopy (OM) and scanning electron microscopy (SEM) images of different zones of obtained hollow xerogel tubes. **d** Illustration of polarized Raman spectroscopy. **e** Homogeneous structures within knots and the wall of obtained xerogel tubes based on Raman spectra acquisition with different angles. Raman spectra were collected at different incident laser polarization. The polar plots refer to the normalized intensity of C-C skeletal stretching vibration at 1105 cm⁻¹ relative to the NH₂ bending vibration at 1625 cm⁻¹. Plastic tubes with an inner diameter of 4.6 mm were used as mold. Unless specified otherwise, all plastic tubes shared the same structural characteristics.

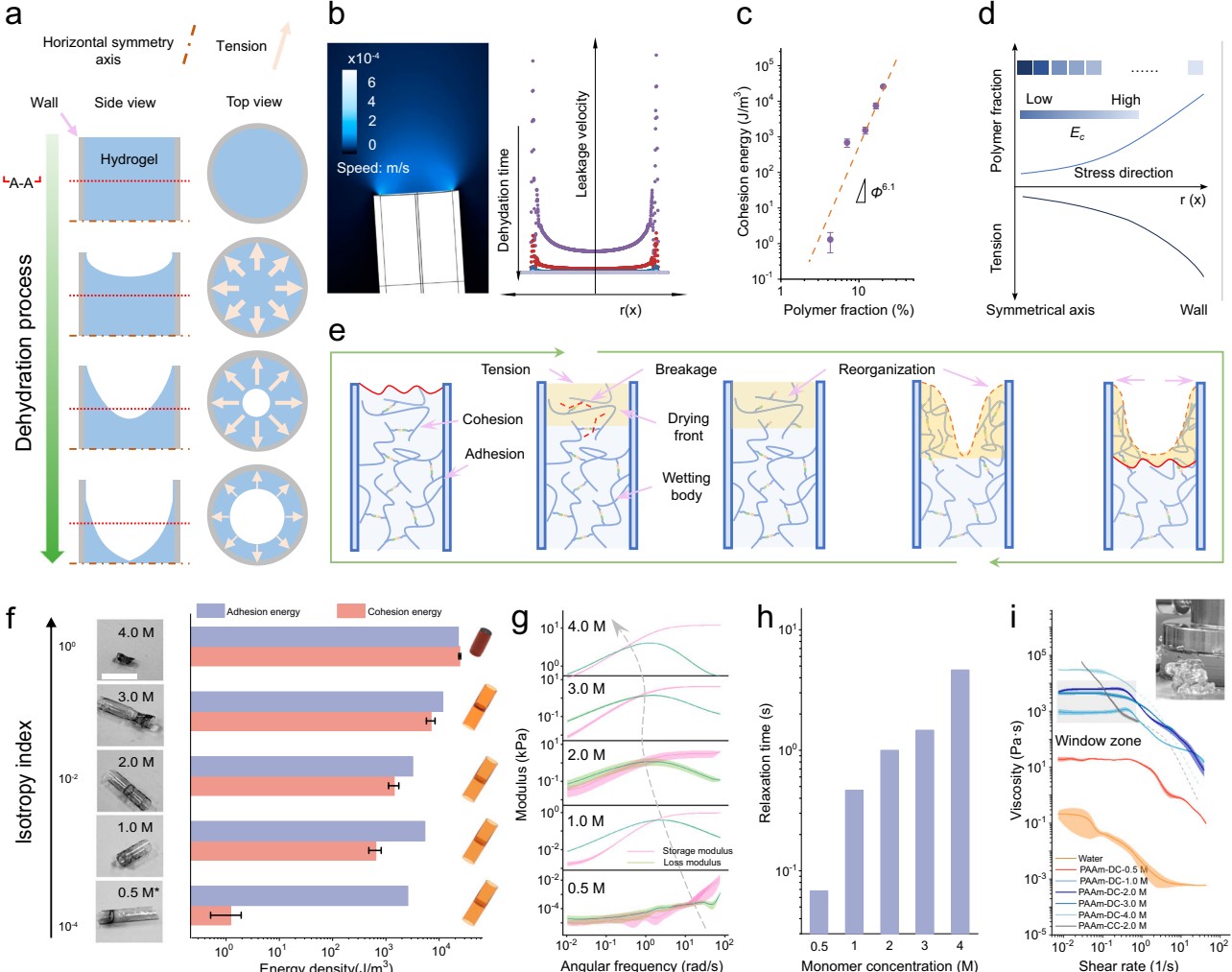

**Fig. 2 | Reorganization of dynamic polymer network shapes the evolution of macrostructure during dehydration. a** The macrostructure evolution of the dehydration process as a result of network tension formation, development and release. **b** A simulated water vapor leakage from hydrogels during the dehydration process. The data was acquired through a COMSOL simulation. **c** The scaling relation between cohesion energies and the polymer fraction. Individual sample numbers N > 5, $\Phi$ refers to the polymer friction. **d** Dehydration-induced network tension. **e** A schematic representation of the micro-level mechanism of dynamic polymer reorganization due to dehydration-induced stress. **f** The dehydration behaviors of dynamic hydrogels and the corresponding cohesion/adhesion energy

of dynamic hydrogels with different monomer concentrations. Insets: the typical final configurations of obtained xerogels with different monomer concentrations. *As for the lower monomer concentration of 0.5 M, the hydrogel will flow out when it is placed vertically in the tube. Therefore, only sealed tubes were positioned vertically. Scale bar is 1 cm. **g** The frequency sweep curves of dynamic hydrogels. **h** Relaxation time of dynamic hydrogels. **i** Viscosities of different hydrogel samples. PAAm stands for polyacrylamide. DC and CC stand for dynamically crosslinked and covalently crosslinked, respectively. The concentrations refer to the amounts of monomers used. The window zone indicates the formation of full coverage by the hollow tubes.

spectroscopy (Fig. 1d). The polar plots were approximately circular for both walls and knots (Fig. 1e), indicating the isotropic organization of polymer chains in the knots or tube wall[20–22]. Their POM images also indicated the isotropy of inner structures. Under crossed polarized light, the color of xerogel wall was independent on the stacking layer of walls (Supplementary Fig. 9). These results revealed the different processes occurring during the generation of hollow xerogel tubes from liquid behavior-assisted organization (Supplementary Figs. 10, 11), e.g. Langmuir–Blodgett method[23] or constant contact radius (CCR) mode[24].

Considering the gradual deformation that takes place during the dehydration process of hydrogels and the dynamics of the network, we propose that the reshaping of dynamic hydrogels is a direct result of the dehydration-induced stress (Fig. 2a). The existence of the tube wall forced a gradient water vapor leakage, and the gradient direction was along the radial direction (Fig. 2b). On the other hand, the scaling law justified the evolution of the cohesion energy within the hydrogels due

to increased polymer fraction ($\Phi$) during dehydration (Fig. 2c, $\Phi^{6.1}$). Therefore, the gradient vapor leakage generated gradient cohesion energy increasing from the center to the boundary of the hydrogel. The internal stress was then created along the radial direction and increased from the center to the boundary (Fig. 2d). During dehydration, the release of inner stress was facilitated by the reshaping of the dynamic polymer network, accompanied by destruction and regeneration of the air-hydrogel interface due to the facile cleavage and rebounding of the dynamic covalent bonds. On the macroscale, in-plate (perpendicular to the direction of movement of the air-hydrogel interface) shrinkage, rather than flowing along the longitudinal axis, dominated the dehydration of the dynamic hydrogels and resulted in xerogel tubes without directional organization (Fig. 2e). The stress perpendicular to the radial direction can be reasonably neglected considering the migration direction of polymers during dehydration. Consequently, a fully covered xerogel tube was formed due to the accumulation of polymer chains migrating from the central zone. In

contrast, the quiescence of networks in covalent hydrogels led to failure of the hydrogel/inner wall interface under the stress induced by dehydration, and later to isotropic shrinkage (Supplementary Figs. 12 and 13).

It is noteworthy that the dehydration modality for the dynamic hydrogels appears to be influenced by the initial parameters, e.g., the monomer concentrations. For the initial hydrogels, the monomer concentrations determine the rheological properties as well as the reorganization behavior of the dynamic polymer network[25]. As the monomer concentration increased, the resulting xerogels evolved from anisotropic hollow tubes to bulk xerogels (Fig. 2f). This result could be induced by the competition between cohesion energy and adhesion energy. For example, in the virginal dynamic hydrogels, as the monomer concentration increased from 0.5 M to 4 M, the cohesion energy exceeded the adhesion energy in the plastic tubes, so that at 4 M monomer the xerogel formed with isotropic shrinkage. In comparison, when the substrate was replaced with glass tubes to enhance the hydrogel-wall interface interaction, the anisotropic hollow tubes appeared again due to the enhanced adhesion energy at the interface (Supplementary Fig. 14). In addition to the competition at the interface, increasing monomer concentration significantly changed the degree of reorganization even within the dynamic network, as reflected by the shift of the crossover point in the frequency sweep curves (Fig. 2g). The relaxation time also increased with increasing monomer concentration, due to more intensive polymer chain entanglement since higher monomer concentration impedes the reorganization of dynamic bonds (Fig. 2h).

Moreover, the failure of the dynamic hydrogel-wall interfaces can also occur due to the sensitivity of the system to the initial characteristics (Supplementary Fig. 15). Firstly, small gaps in the interface may arise due to defects around the wall or during the trimming of the outer tubes, particularly caused by the contrasting moduli of hydrogels and tubes[26,27]. Interfacial interactions are weakened by the formation and development of small gaps, leading to the failure of the hydrogel-wall interface. This, in turn, causes dynamic hydrogels to shrink like covalent hydrogels. In addition, the destruction of the initial hydrogel-wall interface can also result in isotropic shrinkage of dynamic hydrogels. This is likely due to the reduced interface interaction caused by eroded conformality between the virginal hydrogel-wall interface (liquid-solid) and the newly generated hydrogel-wall interface (solid-solid)[28] (Supplementary Fig. 16).

Therefore, the liquid-like properties of dynamic hydrogels, imparted by dynamic bonds, play a critical role in the formation of hollow xerogels for full interface coverage. As shown in Fig. 2i, the zero shear viscosity of the dynamic hydrogels increased with the monomer concentration (the first three values of viscosity were used to extrapolate the value at shear rate = 0, which is considered to be the zero shear viscosity.) (Supplementary Fig. 17). The appropriate viscosity allows the formation of hollow xerogel tubes with a continuous surface (window zone, $10^3$-$10^4$ Pa·s). The dynamic hydrogels of low viscosity (below $10^2$ Pa·s), although the flow behavior is evident on the observed time scale, hollow xerogels still can be generated by the hydrogels in vertically placed plastic molds. In contrast, the non-crosslinked hydrogels, as expected, only formed the discrete, fragmented xerogel adhered to the wall after dehydration due to its low viscosity, resembling the dehydration of liquids (Supplementary Fig. 18).

The dehydration process was systematically assessed in this study by analyzing a variety of intermediate states. In general, the dehydrated hydrogel can be categorized into three regions: the mature zone (zone I) which was nearly identical to the last phases of hollow xerogels, the transition zone (zone II) that shifted along the air-hydrogel interface, and the wetting zone (zone III) possessing features typical of a hydrogel (Fig. 3a). Upon the onset of dehydration, zone I emerged and developed. Meanwhile, zone III transitioned to zone I and zone II, as the water content decreased. After freezing in liquid

nitrogen, zone I displayed transparency due to low water content and less formation of ice crystals, while zones II and III became opaque due to the formation of large quantities of ice crystals[29] (Supplementary Fig. 19). Furthermore, micro-sized porous structures emerged due to the loss of water and the reorganization of the dynamic network (Fig. 3b). The region surrounding the air-hydrogel interface possessed a higher density, with considerably smaller pores compared to those present in the virginal hydrogels (Supplementary Fig. 20). A gradient distribution of pore sizes was observed within zone II and was primarily attributed to the migration and reorganization of the polymer networks (Supplementary Fig. 21). Only submicron pores (50–60 nm) were found in zone I or in the xerogel tubes that eventually formed.

The resulting macrostructure of hollow xerogel tubes can be regulated by modulating the initial parameters. When a fresh-cut surface was produced manually, the knot position in the final xerogel tubes showed a tendency to move towards the existing original surface rather than the midpoint of the longitudinal direction for the dynamic hydrogel without intervention. Specifically, only a dense, dead zone formed quickly around the initial air-hydrogel interface, while the porous active zone from the fresh-cut surface facilitated the steady migration of the air-hydrogel interface (Supplementary Fig. 22). Finally, the knot was secured near the exiting initial air-hydrogel interface, away from the horizontal symmetry axis of xerogels. Therefore, the position of the knot can be regulated by the relative activity of the air-hydrogel interface. Compared to those with the knot in the middle, the porous microstructures of xerogels with changing knot positions remained, irrespective of the treatment, to generate a fresh-cut surface (Fig. 3c). Given the intrinsic properties of dynamic hydrogels, the fresh-cut surface became more active by creating free sites in polymer networks while separating polymer chains[30]. This made it easier for the dynamic networks to reorganize around the fresh-cut surface compared to the original air-hydrogel interface. Interestingly, the dead zone emerged around the end with the initial air-hydrogel interface in the presence of a highly active interface. Such a phenomenon can be induced by the difference in the air-hydrogel interfaces around the two ends. In other words, the water within the dynamic hydrogels tended to leak through the fresh-cut surface, in contrast to the initial air-hydrogel interface. Thus, the accumulation of polymers within the plane can be enhanced by modulation of the initial air-hydrogel interface due to the impaired hydroplastification. As a result, an excess accumulation of polymers within the plane (perpendicular to the migration of the air-hydrogel interface) appeared as dead zone. This finding aligns with prior research indicating that the mobility of polymer chains within hydrogels may impact the diffusion of water molecules in the network, ultimately determining the overall rate of evaporation[8,31]. Higher polymer chain mobility typically results in increased water diffusion and faster evaporation rates, while lower mobility can result in slower evaporation[32].

By introducing the fresh-cut surface, four distinct modes of hydrogel dehydration in cylindrical vessels can be generated. These modes are defined primarily by their initial parameters/conditions (Fig. 3d). For the hydrogels with both interfaces as the virginal air-hydrogel interfaces or fresh-cut surfaces, the knots were located in the center regions of the resulting hollow xerogels. For the tubes with one end sealed, the dehydration of the dynamic hydrogels inside took place starting from the open end, so that the knots move toward the sealed end with a dead zone. Generating one fresh-cut surface proved analogous behavior to that of closed end mode wherein the knot appeared towards the exiting initial end. These findings indicate that the location of the knot is primarily determined by the rate of water vapor leakage competing at both ends. The thickness of the knots, following dynamic dehydration of the hydrogels in various modes, was comparable to that of the walls. Therefore, the reshaping of the dynamic hydrogels was verified to be dominated by the in-plate migration of the dynamic polymer networks induced by the stress

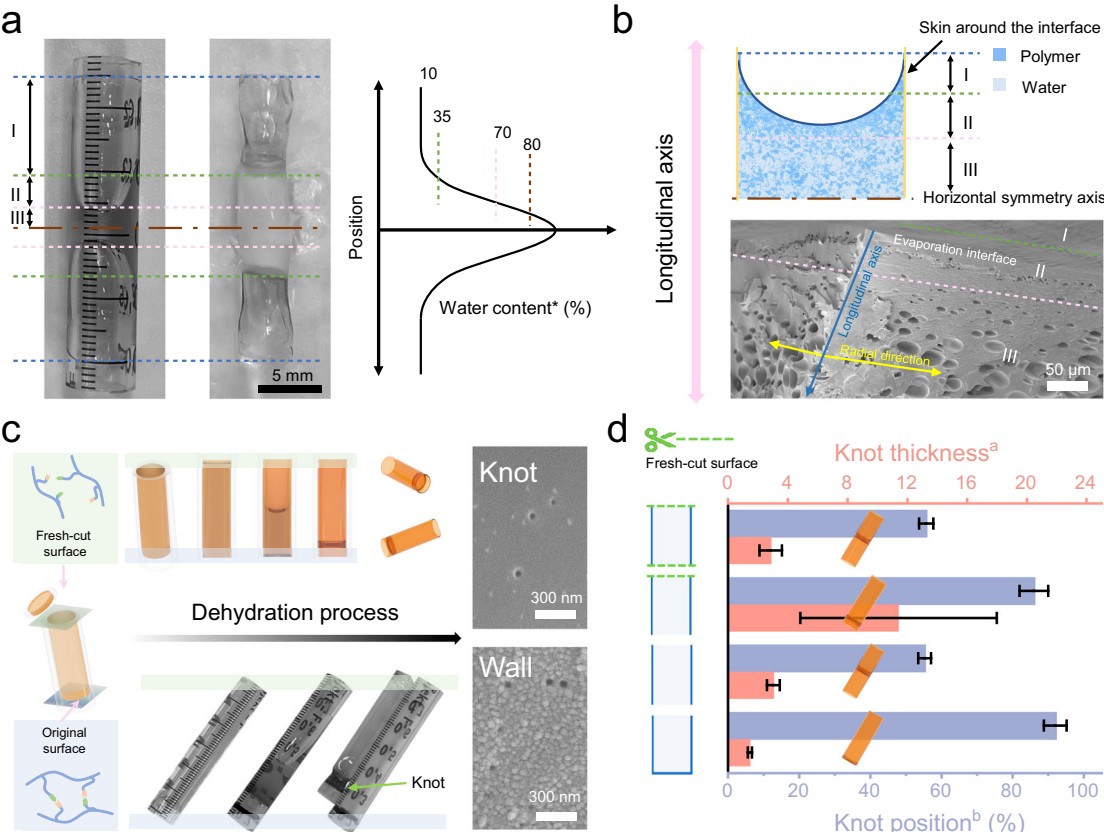

**Fig. 3 | Evolution of structural features during the dehydration of hydrogels.** **a** Photos depicting the intermediate status during dehydration. The left image is of the sample without treatment and the right image is of the sample after liquid nitrogen treatment. The plot on the right is a graphical representation of water contents with respect to vertical position in the tube. * In the intermediate state, water content is determined by weight. **b** Structural evolution of hydrogels during dehydration. **c** A cleaving method used to manipulate the knot position. **d** A graphical representation of the dependence of the knot position as a result of the cleaving position. [a]Knot thickness is normalized to wall thickness to avoid discrepancies between systems. [b] Knot position is defined as the ratio of the maximum length between the knot and any end of the hollow xerogels to the total length of the xerogels.

along the radial direction, rather than the accumulated stress along the longitudinal direction.

Anisotropic dehydration, which relies on dynamic polymer networks, was found to be widespread among various materials (Fig. 4). For instance, dynamic hydrogels combined with a binary solvent system still formed the hollow xerogel tubes (Fig. 4a, b), despite variations in the speed of evaporation. Even for the system consisting of multi-layered hydrogels, the dehydration-induced stress helped the reorganization of dynamic networks (Fig. 4c, the addition of dyes resulted in the color for clearly distinguishing diverse layers). A hollow xerogel tube with an internal knot was created with a localized and evenly distributed color similar to the original system. The knot formed in the middle zone of the multilayered hydrogel. These results indicated that the migration of polymers within the plane determines the dehydration. The interfaces between different layers were penetrated due to the reorganization of dynamic bonds and the entanglement of polymer chains[33,34]. Thus, the interfaces within the different dynamic hydrogel layers allowed rapid reorganization during dehydration. The primary knot was only formed in the central zone, while the intrinsic interfaces are visible in the resulting xerogel tubes.

The hollow xerogel tubes were also constructed by different monomers, such as positively charged monomers (Figs. 4d, 2-(dimethylamino)ethylacrylate methyl chloride quaternary salt), negatively charged monomers (Fig. 4e, sodium p-styrenesulfonate) or other neutral monomers (Supplementary Fig. 23, 2-hydroxyethyl methacrylate). The resulting xerogel tubes formed by these dynamic hydrogels further confirmed the general principles of the polymer network reorganization within dynamic hydrogels during dehydration.

Such reorganization of dynamic hydrogels during dehydration has also been found in other dynamic bonding systems, e.g., with a Schiff base as crosslinkers (Fig. 4f). The resulting xerogel tubes had similar structural characteristics as those with the boronate ester as crosslinkers. Compared to crosslinked networks with boronate esters, xerogels crosslinked with a Schiff base contained failure at the hydrogel-wall interface in the vicinity of the knot, which behaved as the shrunken and curved surface. This could be induced by the sharp fluctuation of the monomer fraction around the knot in the late stage of dehydration and the longer relaxation time scale of dynamic crosslink sites (Supplementary Fig. 24). Therefore, the fast reorganization of dynamic hydrogels during dehydration predominantly contributed to the transition of xerogels from isotropy to anisotropy, while a fast dissociation of the crosslinkers plays a critical role (Fig. 2).

The geometric parameters had a limited influence on such unique dehydration modality (Supplementary Fig. 25). A cone-shaped xerogel was generated using the appropriate model. Additionally, the diameter of the hollow xerogel tubes may surpass 1 cm, with no visible accumulation of thickness around the knots. It was also found that the different xerogel wall thicknesses could be achieved in different systems by adjusting the diameter and polymer fraction, because the generation of xerogel wall was contingent upon the accumulation of polymers along the radial direction (Supplementary Fig. 26). Therefore, the dehydration-induced stress, which drives polymer migration,

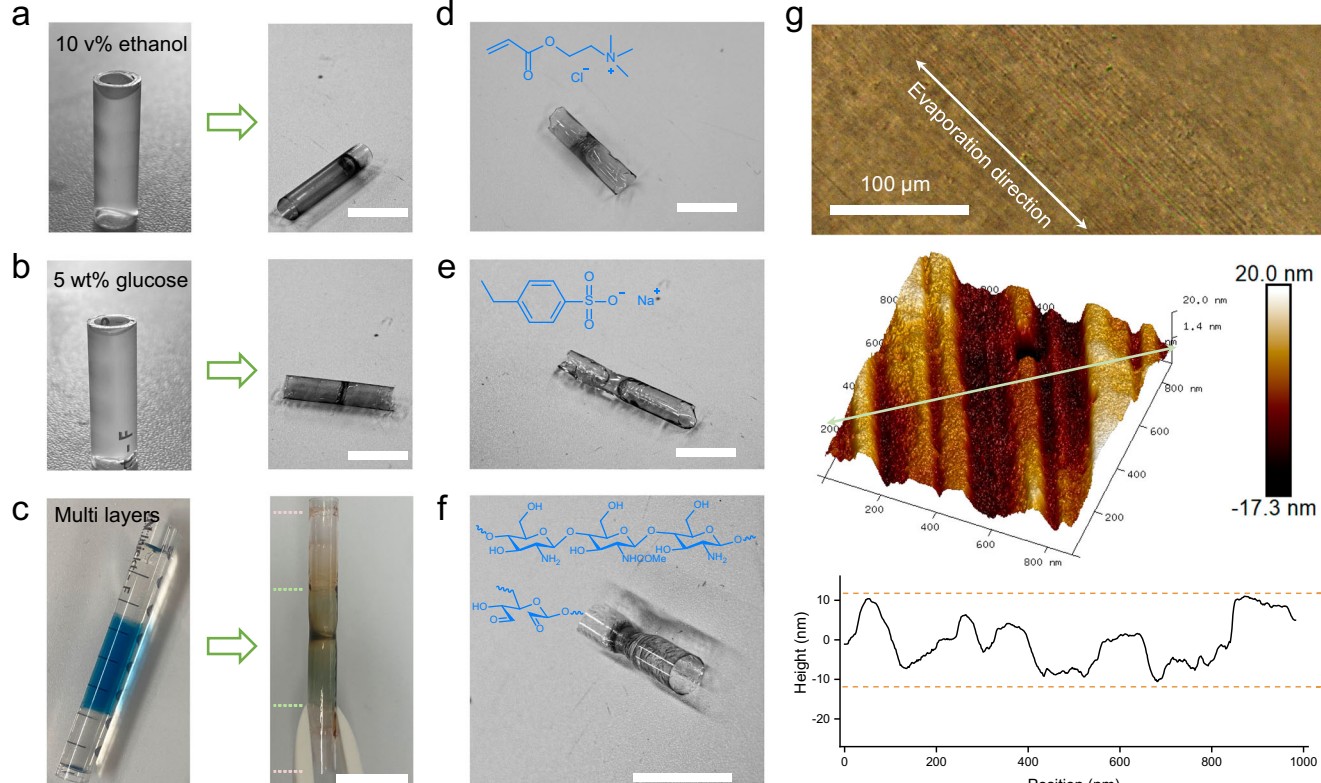

**Fig. 4 | Dynamic reorganization is universal for dynamic hydrogels during dehydration. a** Photos of a dynamic hydrogel using binary solvent system consisting of 10 v% ethanol in water and the resulting xerogel. **b** Photos of dynamic hydrogel in an aqueous solution consisting of 5 wt% glucose and the resulting xerogel. **c** Photos of the triple-layered hydrogel and the resulting xerogel. The interfaces between different tiers have been penetrated. **d** A photo of the resulting xerogel prepared via the dehydration of a dynamic hydrogel composed of positively charged monomers, here with 2-(dimethylamino)ethylacrylate methyl chloride quaternary salt as a representative example. Note that under ambient conditions, the xerogel tubes retain viscous and soft properties for three months.

To extract it, the samples were treated in an oven at 60 °C for 8 hours. **e** A photo of the xerogel prepared by dehydration of a dynamic hydrogel consisting of negatively charged monomers, here with sodium p-styrenesulfonate as example. **f** A photo of the xerogel resulting from the dehydration of a dynamic hydrogel crosslinked by a Schiff base between amine and aldehyde, e.g., with oxidized dextran and chitosan. Scale bars are 1 cm. **g** The resulting xerogel with a high-resolution imaging of the nanostructure after dehydration. The AFM images were taken on the outer surface of hollow xerogel tubes with periodic trenches. The height data was acquired along the green line in the AFM image.

proved to be an effective approach even on a large scale, with broad applicability.

Creating nanopatterns on a large-scale surface is often challenging and usually requires complex devices or materials[35–37], especially for curved surfaces. Taking advantage of the good conformality of the dynamic hydrogels within the molds, fine structures on the inner walls were transferred onto the outer surface of the resulting xerogels after dehydration. For the tubes with a series of trenches on the inner wall (Supplementary Fig. 27), such patterns were engraved around the hydrogel-wall interfaces (Supplementary Fig. 16a) and were maintained until the end of dehydration (Fig. 4g). The resolution of patterns in the curved surface in the vertical direction was up to sub-10 nm. By contrast, no patterns were generated on the xerogel surface when the smooth substrate was utilized (Supplementary Fig. 28). The nanostructures created on a 3D curved surface on a large scale by anisotropic dehydration of dynamic hydrogels were demonstrated, e.g., the length of hollow xerogel tubes with nanopatterns could be up to ~10 cm, while the diameter could be up to ~1 cm (Supplementary Fig. 29). Therefore, the external surface of the centimeter-scale hollow xerogels can be used to complementarily engrave structural features of the curved substrates. Given this, the anisotropic dehydration of dynamic hydrogels has great potential in various applications for constructing nanopatterns on curved surfaces on a large scale.

## Discussion

In summary, the reorganization of dynamic hydrogels during dehydration was systematically investigated. Unlike the typical isotropic contraction of covalent hydrogels during dehydration, a fast reorganization of dynamic hydrogels facilitated the migration of polymer chains under the dehydration-induced stress. Hollow xerogel tubes with unique wall-knot structures can be generated from dynamic hydrogels, based on a criterion that considers the competition between cohesion energy and adhesion energy. These tubes can be used to engrave nanopatterns onto a substrate through macroscopic structural reconstruction. Meanwhile, such reorganization during dehydration is demonstrated to be universal for dynamic hydrogels, as confirmed by using various monomers and solvents. The structure of xerogel tubes, such as wall thickness and knot position, could also be regulated by dynamic bonding and dehydration process. Combined with the conformality of hydrogel-wall interfaces, surface nanostructures of substrates can be engraved on resulting hollow xerogel tubes, which allows the large-scale fabrication of curved surfaces with nanopatterns. Therefore, we offer a different and widely applicable perspective to comprehend the inherent characteristics of dynamic hydrogels. The potential applications of large-scale curved surfaces produced through universal anisotropic dehydration with excellent conformality are significant, such as in generating metasurface and for nanotransfer printing.

## Methods

### Materials

Acrylamide was purchased from MP biomedicals LLC (France). N, N′-Methylenebisacrylamide (MBAA), lithium phenyl−2,4,6-trimethylbenzoylphosphinate (LAP), dextran (Dex, $M_r$-70000), diethylene glycol (DEG), chitosan, sodium periodate and phenylboronate acrylamide (PBAAm) were bought from Sigma-Aldrich (Germany). Dopamine methylacrylate (DMA) was purchased for TCI (Germany). Borax-NaOH buffer (pH = 10.00) was purchased from TH Geyer (Germany). DI water was used in all steps and all solvents were used directly from TH Geyer (Germany) without further treatment.

### Preparation of hydrogels

Unless otherwise stated, dynamic hydrogels refer to boronate ester crosslinked hydrogels. Usually, the precursor of hydrogels consists of 2 M acrylamide (142 mg/mL), 1.25 mol% PBAAm/DMA (based on acrylamide) and buffer solution. After adding 0.5 wt% LAP into the hydrogel precursor solutions under sonication condition, photoinitiated radical polymerization was carried out with a UV source ($\lambda$ = 356 nm, irradiance is 1.8 mW/cm$^2$) for 10 min. Specifically, the precursors with LAP were transferred into the model before polymerization. The disposable injectors were used as cylindrical plastic tubes to mold hydrogels (TH. Geyer, Germany). A hydrophilic inner surface was also used by using a common NMR tube (TH. Geyer, Germany, inner diameter 4.19 ± 0.05 mm).

Another type of dynamic hydrogel based on Schiff base consists of oxidized dextran (Odex), chitosan, and acrylamide. First, the chitosan solution was prepared by dissolving a certain amount of chitosan in 2.1% (w/v) acetic acid aqueous solution and stirring overnight. The dynamic Schiff's base hydrogels were then prepared by mixing the Odex aqueous solution, chitosan solution, acrylamide, MBAA, and LAP with DI water, followed by photoinitiated polymerization[38]. Here, Odex was synthesized according to reported works[39,40]. Typically, 5 g of dextran was dissolved in 200 mL of deionized water, followed by the addition of 100 mL of an aqueous solution of NaIO$_4$ (containing 4 g of NaIO$_4$). The reaction was conducted under N$_2$ atmosphere, shielded from light, with stirring at room temperature for 24 hours. Subsequently, an equimolar amount of diethylene glycol was added to quench the unreacted NaIO$_4$. The resulting solution was then subjected to dialysis for three days against DI water (molecular weight cut off is 3500). The pure Odex was obtained through lyophilization. The chemical structure of Odex was further confirmed by FT-IR and Raman analysis (Supplementary Fig. 30).

Covalently crosslinked hydrogels with a similar crosslink density (0.025 mol/L) were prepared by using MBAA as crosslinker, while the other compositions were the same.

The hydrogels without crosslinking were prepared using the same amount of the same components except adding crosslinker. The hydrogels were viscous and appeared to flow within observation time.

### Dehydration of hydrogels

Prepared hydrogels with outer tubes were left in ambient conditions until completely dry. Normally, the tubes containing the hydrogels would be placed in a vertical position during dehydration. For the dynamic hydrogels that are of low viscosity and flow behavior within observation time, the tubes with a single sealed end will be applied to avoid hydrogel spillage. For the hydrogels that would not flow out freely within observation time, we found that the dehydration process was independent of the position of the plastic molds (placed in vertical position or horizontal position). The influence of the substrates on the density of resulting xerogels is not evident based on the statistical analysis (Supplementary Fig. 31). A fresh-cut surface was generated via using disposable blades (Leica 819) by hand. In the typical procedure, a thin metal rod is employed to gradually dislodge the xerogels from the tubes by pushing the knot after the hydrogel has been dehydrated. The

difference between inside diameter of the tube and the outside diameter of the hollow xerogel tube was negligible (Supplementary Fig. 32).

### Generation of xerogels with nanopatterns on the surface

To obtain the xerogels with nanopatterns on the surface, a cylindrical plastic tube (1 mL disposable syringe Injekt® F, Carl Roth, Germany) with an internal diameter of 4.60 mm was used as the model for the molding of the hydrogels. As the commercial products, the inner surface of the tube with a series of trenches was identified (Supplementary Fig. 27). Following dehydration, the dynamic hydrogels that had formed in situ within the plastic tubes underwent a transformation, becoming hollow xerogel tubes. The nanopatterns were observed on the outer surface of the xerogel tubes, which corresponded to the inner surface of the plastic tubes.

### Evaluation of cohesion energy, adhesion energy, and the scaling law

The cohesion energy was calculated based on the analysis of the linear viscoelastic regions (Supplementary Note 1). The adhesion energy was calculated based on the displacement-force curves (Supplementary Note 2). A scaling law was used as the extrapolation principle (Supplementary Note 3).

### Simulation of hydrogel dehydration in tubes

COMSOL Multiphysics® software was utilized to simulate the dehydration behavior of hydrogels. The Stefan flow was considered. The geometric parameters were set according to those in the experiments (Supplementary Note 5).

### Characterization

The polarized Raman spectra were recorded via LabRAM HR Evolution (Horiba France SAS) system. A 532 nm laser was applied with the ND filter (3.2 % transmission). The acquisition time was 10 seconds, and each individual spectrum was the accumulation of 4 scans. The mechanical properties were measured using a Z3 micro tensile test machine, which was equipped with a 50 N load cell. The rheological properties of hybrid hydrogels were tested using an HR 20 rheometer (TA Instruments, USA) with the UV accessory (OmniCure, Germany) (Illustrated in Supplementary Fig. 33). A parallel plate with 20.00 mm diameter was applied and the distance of gap was set to 300 μm. An LEO supra-35 high-resolution field emission scanning electron microscope (Carl Zeiss AG, Germany) was used to characterize the microstructure of various samples and the targeted voltage was 5 kV. The optical and polarized images of samples were collected using an Eclipse 600 microscope from Nikon. Fourier Transform Infrared Spectroscopy (FT-IR) was recorded on Alpha FT-IR Spectrometer (Bruker, Germany) at room temperature. The samples were measured between 4000 and 400 cm$^{-1}$ with a resolution of 2 cm$^{-1}$ using Platinum ATR and accumulated 32 scans. A DSA25 Drop Shape Analyzer (KRÜSS, Germany) was applied to measure the static water contact angle on various substrates. The method of sessile drop was used and the images were recorded were analyzed[41,42]. The volume of drop was set as 4 μL under the rate of 2.67 μL/s. Atomic force microscopy (AFM) characterization was performed in the dry state using a Multimode 8 AFM (Bruker, Karlsruhe, Germany) with a NanoScope V controller in an ambient environment. Soft probes (ScanAsyst-Air, Bruker) with a nominal spring constant of 0.4 N·m$^{-1}$, and a nominal resonance frequency of 70 kHz were used. The ScanAsyst-air mode was used as the imaging mode, with a scan rate of 0.97 Hz and a resolution of 512 × 512 pixels per line.

## Data availability

The data generated in this study are provided in the Supplementary Information/Source Data file. All other relevant data supporting the findings of this study are available from the corresponding authors upon request. Source data are provided with this paper.

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

## Acknowledgements

K.Z. thanks the German Research Foundation (DFG) and Lower Saxony Ministry of Science and Culture for the project INST186/1281-1/FUGG. K.Z. thanks the EU for finally supporting the project 'NEuM' (ZW7-85191973). D.X., X.M. and S.Y. thank the China Scholarship Council (CSC) for the financial support of their PhD study. J.P. is grateful for Alexander von Humboldt and National Science and Engineering Research Council (NSERC) postdoctoral fellowships. The authors thank Dr. Florian Ehlers from the Institute of Physical Chemistry, University of Göttingen, for providing device support in AFM characterization.

## Author contributions

K.Z. developed the concept. D.X. performed most of the experiments, analyzed the experimental data and wrote the paper. X.M. and P.V.

supported AFM characterization and data analysis. S.L. helped to perform mechanical characterization. J.P. helped to organize and optimize the Figures. All authors discussed the results and reviewed the manuscript.

## Funding

## Competing interests
The authors declare no competing interests.
