## [Peer Review File · Nature Communications]

Dehydration regulates structural reorganization of dynamic hydrogelsEditorial Note: Parts of this Peer Review File have been redacted as indicated to remove third-party material where no permission to publish could be obtained.

REVIEWER COMMENTS

Reviewer #1 (Remarks to the Author):

My comments to the Authors are in the file Review.pdf

[Editorial Note: This PDF is displayed after Reviewer #4's comments]

Reviewer #2 (Remarks to the Author):

Reviewer #3 (Remarks to the Author):

The manuscript by Xu et al. on the effects of dehydration on the structural reorganization of dynamic hydrogels covers an interesting and emerging topic in the community. The authors specifically study the mechanical deformation and structural rearrangements observed in boronate ester hydrogels undergoing dehydration. While the topic and many of the observations are interesting, the manuscript is rather dense and obtuse in phrasing and it is difficult to extract the main conclusions or hypotheses from the document in its current form. As such, it is premature for publication in Nature Communications. However, given the interest in this topic and its relatively unexplored nature, the authors should be encouraged to restructure and reformulate their work to make the following points more clear.

1. What are the central hypotheses and conclusions of the work in simple, accessible language?
2. What are the broader implications of these hypotheses and conclusions? What excites a general audience as for Nature Communications as opposed to a more focused polymer processing community?
3. How do the underlying physics of the system relate to the dynamics of the system? This is alluded to but not studied clearly and in detail and this is the main advantage of using a boronate ester-based system as it is highly tunable in its dynamics.
4. Finally, the manuscript text and figures should be clarified to be accessible to a general

audience. Here, less is often more, and making sure that the central points are communicated clearly is more powerful than an overwhelming amount of information and at times overly vague text.

Reviewer #4 (Remarks to the Author):

In this work, the authors investigate the dehydration behaviors of covalent or dynamic cross-linked hydrogels filled in hollow tubes. They revealed that the dynamic cross-linked gels dehydrated to give the xerogels unique wall-knot structures by reorganization of the cross-linked network when their adhesion energy with the surface is higher than their cohesion energy. The authors well investigated the validity of their works through rheological measurements, structural investigations, and simulation. However, there is still ambiguity in the experimental details. Some concerns ought to be given attention, as follows:

Comment 1:

I would like to know how to take out the xerogels with wall-knot structures from the hollow tubes. Is it possible for the xerogels with wall-knot structures to detach from the surface of the tube due to the cohesive energy exceeding the adhesive energy by the increase in concentration during the drying process?

Comment 2:

Some xerogels have narrow parts near the knit while other xerogels do not show the narrow parts (ex. Figure 2f). What factors determine which structures are obtained?

Comment 3:

The authors should show the details of how to obtain the xerogel surface with the aligned sub-10 nm patterns. Is it possible for similar methods to be applied to the normal hydrogels or xerogels without hollow tubes? That is important to consider whether this method is novel.

Comment 4:

The hollow tubes with various diameters were used in this work. What effect does the diameter have on the structure obtained?

Comment 5:

In Supplementary Figure 4, is it possible to compare water contact angles on surfaces with different curvatures, appropriately?

Review of Manuscript NCOMMS-24-07416-T

Dehydration regulates structural reorganization of dynamic hydrogels

Dan Xu, Xintong Meng, Siyuan Liu, Jade Poisson, Philipp Vana and Kai Zhang

In this paper, the Authors report on the anisotropic dehydration modality of dynamic hydrogels and highlight the reorganization of polymer networks within dynamic hydrogels, triggered by dehydration-induced stress.

The goal of the paper is to provide a novel and precise description of a dehydration-induced gradient stress distribution for dynamic hydrogel system.

Review

The topic of the article is interesting. The Authors showed with great details what they aimed to show: the differences in the dehydration modality between static (covalently cross-linked) and dynamics hydrogels. Moreover, they also investigated how the hydrogel viscoelasticity affect the dehydration process.

However, in my opinion, the article can be greatly improved in many of its aspects, as I list below. So, I recommend a **major revision**.

Major issues

1. In general, the manuscript is quite dense and the figures have a very important role in helping the reader to follow and understand the text. This is why I would ask the Authors to improve them.
 - In Figure 1 the current sizes "t" and "l" used for the anisotropic case are confusing, and a change in the symbol used to denote current radius and length would be appropriate.
 - Figure 2g seems a very important figure. In my opinion, it could be improved by showing also the result in terms of the final configuration achieved with different monomer concentrations.
2. Theoretical and modeling results are not appropriately illustrated in the paper; it would be interesting to know which components of the stress are important during the dehydration process; can the Authors say something about it?

3. The authors indicate at the end of the article that a Simulation of hydrogel dehydration in tube was performed with COMSOL and a Stefan flow. It is not clear where these results are showed in the article.
4. Moreover, which model did they use? Can they write the equations with boundary and initial conditions in the supplementary material?

Finally, English language as well as several typos must be improved and corrected.

Reviewer #1 & #2 (Remarks to the Author):

[Reviewer #2: I co-reviewed this manuscript with one of the reviewers who provided the listed reports. This is part of the Nature Communications initiative to facilitate training in peer review and to provide appropriate recognition for Early Career Researchers who co-review manuscripts.]

In this paper, the Authors report on the anisotropic dehydration modality of dynamic hydrogels and highlight the reorganization of polymer networks within dynamic hydrogels, triggered by dehydration-induced stress.

The goal of the paper is to provide a novel and precise description of a dehydration-induced gradient stress distribution for dynamic hydrogel system.

The topic of the article is interesting. The Authors showed with great details what they aimed to show: the differences in the dehydration modality between static (covalently cross-linked) and dynamics hydrogels. Moreover, they also investigated how the hydrogel viscoelasticity affect the dehydration process.

However, in my opinion, the article can be greatly improved in many of its aspects, as I list below. So, I recommend a major revision.

Response: We thank the reviewer for the very positive comments on our research work. We are pleased to learn that the reviewer fully recognized the importance and novelty of this study. Below, we have provided a point-by-point response to the valuable comments.

1. In general, the manuscript is quite dense and the figures have a very important role in helping the reader to follow and understand the text. This is why I would ask the Authors to improve them.

· In Figure 1 the current sizes "t" and "l" used for the anisotropic case are confusing, and a change in the symbol used to denote current radius and length would be appropriate.

· Figure 2g seems a very important figure. In my opinion, it could be improved by showing also the result in terms of the final configuration achieved with different monomer concentrations.

Response: We appreciate the reviewer's suggestions and insightful comments. We apologize for any misleading descriptions or information due to improper demonstration. We further improved Figure 1a and Figure 2g to demonstrate the results more clearly and precisely.

- In Figure 1, the symbols refer to the radius and length has been replaced. Specifically, "t" was replaced with "D" and "l" was replaced with "L₁". The corresponding text in the manuscript has also been revised to avoid any misleading descriptions.

Fig 1a. Two typical dehydration behaviors of hydrogels. Type I: covalent hydrogels with isotropic dehydration process; Type II: dynamic hydrogels with anisotropic dehydration process.

- We appreciate the reviewer’s thorough understanding about the current work and Figure 2g is quite important in the context. The experimental results and theoretical analysis confirm the isotropic organization of polymer chains. Therefore, we mainly discuss the macrostructure evolution of hydrogels during the dehydration process. The final configuration achieved with different monomer concentrations can be found in Figure 2f, along with the corresponding isotropy index. With the increasing monomer concentrations, the dehydration modalities of samples changed from anisotropic hollow tubes to bulk xerogels. Such description was added in the manuscript on Page 8-9:

“As the monomer concentration increased, the resulting xerogels evolved from anisotropic hollow tubes to bulk xerogels (Figure 2f)”

For better understanding, we have merged Fig 2f and g of the previous version. In the current Fig. 2f, the final configuration of the resulting xerogels has been included.

Fig 2f. The dehydration behaviors of dynamic hydrogels and the corresponding cohesion/adhesion energy of dynamic hydrogels with different monomer concentrations. Insets: the typical final configurations of obtained xerogels with different monomer concentrations. *As for the lower monomer concentration of 0.5 M, the hydrogel will flow out when it is placed vertically in the tube. Therefore, only sealed tubes were positioned vertically. Scale bar is 1 cm.

2. Theoretical and modeling results are not appropriately illustrated in the paper; it would be interesting to know which components of the stress are important during the dehydration process; can the Authors say something about it?

Response: Thanks for the kind comments and suggestions. We apologies for a typo in the manuscript on Page 8 and any misunderstanding this may have caused. The original description “The existence of the tube wall forced a gradient water vapor leakage, and the gradient direction was perpendicular to the radial direction” has been revised to “The existence of the tube wall forced a gradient water vapor leakage, and the gradient direction was along the radial direction”. These theoretical results correspond well with the experimental phenomenon.

As for the stress during hydrogel dehydration, it can be considered in two parts: one along the radial direction and the other perpendicular to the radial direction. In this manuscript, we mainly discussed the internal stress along the radial direction based on theoretical analysis and experimental phenomenon. Such description can also be found in the manuscript on Page 8:

“On the macroscale, *in-plate* (perpendicular to the direction of movement of the air-hydrogel interface) shrinkage, rather than flowing along the longitudinal axis, dominated the dehydration of the dynamic hydrogels and resulted in xerogel tubes without directional organization (Figure 2e).”

Based on these experimental results, the stress perpendicular to the radial direction can be reasonably ignored. This is because it cannot cause in-plate migration of polymer chains. The hydrogel-air interface moves quite slowly (less than 10^{-7} m/s) during dehydration (Supplementary Figure 37), indicating that the stress perpendicular to the radial direction is very weak. In addition, the hollow xerogel tubes obtained are independent of whether the samples are placed vertically or horizontally. Finally, it is estimated that the Reynolds number in the system is so low (less than 4.6×10^{-11}) that it cannot induce a boundary layer separation. In other words, the stress perpendicular to the radial direction cannot generate such hollow tubes in view of fluid mechanics if the dehydration process is dominated by the stress perpendicular to the radial direction. The similar thickness and density of the wall and knots, which are the phenomena observed, contradict the idea of stress perpendicular to the radial direction being dominant. Therefore, the stress perpendicular to the radial direction can be neglected in the current manuscript. We also added the detailed descriptions in the manuscript on Page 8 and 14:

“The stress perpendicular to the radial direction can be reasonably neglected considering the migration direction of polymers during dehydration.”

“Therefore, the reshaping of the dynamic hydrogels was verified to be dominated by the in-plate migration of the dynamic polymer networks induced by the stress along the radial direction, rather than the accumulated stress along the longitudinal direction.”

3. The authors indicate at the end of the article that a Simulation of hydrogel dehydration in tube was performed with COMSOL and a Stefan flow. It is not clear where these results are showed in the article.

Response: We thank the reviewer for the suggestive comments. In the current manuscript, the data provided in Figure 2b was acquired via COMSOL based on a Stefan flow. Such description can also be found in the Figure Legend 2b on Page 7:

“A simulated water vapor leakage from hydrogels during the dehydration process. The data was acquired through a COMSOL simulation.”

Detailed information about the simulation can be found in the Supplementary Information (Supplementary Note 5).

4. Moreover, which model did they use? Can they write the equations with boundary and initial conditions in the supplementary material?

Response: Thank you for attentive advice. We have added the detailed information in the Supplementary Information (Supplementary Note 5). Such description can also be found in Supplementary Information from Page 40 to 42:

“Vapor molecules emitted from the hydrogel surface diffuse into the air, which is called as vapor diffusion as seen in Supplementary Figure 47. Similarly, air molecules diffuse towards the hydrogel surface due to the concentration gradient, a process known as air diffusion (as seen in Supplementary Figure 47). During evaporation, air molecules transport downwards due to the presence of a concentration gradient. As the hydrogel surface-vapor interface is impermeable to air molecules, an upward Stefan flow forms to maintain the vapor pressure at the interface. The convection of air enhanced the dehydration process.

Supplementary Figure 47. Interfacial mass transport mechanisms during hydrogel dehydration near hydrogel surface.

First, the diffusion flow of air must be balanced by a convective flow in opposite direction:

$$\Delta m_{air,diff} = -\Delta m_{air,conv} \quad (13)$$

Where Δm is the local mass flux at the surface due to the phase change.

Considering mass flow in a steady state:

$$\Delta m_{air} = \Delta m_{air,diff} + \Delta m_{air,conv} = q_{air} + \rho v \xi_{air} \quad (14)$$

$$\Delta m_{vapor} = \Delta m_{vapor,diff} + \Delta m_{vapor,conv} = q_{vapor} + \rho v \xi_{vapor} \quad (15)$$

Where q is the local mass flux induced by diffusion.

Equation. 13 and **Equation. 14** can be added together to yield

$$\Delta m_{air} + \Delta m_{vapor} = (q_{air} + q_{vapor}) + \rho v (\xi_{air} + \xi_{vapor}) \quad (16)$$

Here,

$$\xi_{air} + \xi_{vapor} = 1 \quad (17)$$

Where ξ is the mass fraction of species.

Taking account of the steady state,

$$q_{air} + q_{vapor} = 0 \quad (18)$$

Substituting **Equation. 17**, **Equation. 18** into **Equation. 16**,

$$\Delta m_{vapor} = \rho v = \Delta m_{total} = \Delta m \quad (19)$$

Here, v is the Stefan velocity that we discussed.

In COMSOL, the evaporation flux is defined as,

$$\Delta m = \begin{cases} M_v K (c_v - c_{sat}), & \text{if } c_v > c_{sat} \text{ or } c_l > 0 \\ 0 & \text{otherwise} \end{cases} \quad (20)$$

Where M_v is the molar mass of water vapor, K is the evaporation rate factor, c_{sat} is the saturation concentration of the vapor, c_v is the vapor concentration and c_l is the liquid water concentration on surface.

The liquid water concentration on surface is computed by solving the following equation,

$$M_v \frac{\partial c_l}{\partial t} = -\Delta m \quad (21)$$

with the initial condition:

$$c_l(0) = c_{l,int} \quad (22)$$

The latent heat source J_{evap} is obtained by multiplying the evaporation flux by the latent heat of evaporation L_v :

$$J_{evap} = L_v \Delta m \quad (23)$$

Due to the endothermic nature of the dehydration process and the relatively higher thermal conductivity of the tube wall, a gradient Stefan velocity distribution along the radial direction is reasonably observed.

The boundary conditions are illustrated in Supplementary Figure 48. Generally,

$$u = 0, v = v_0, T = T_{amb} \text{ at } y = 0 \quad (24)$$

$$u = 0, T = T_{amb} \text{ at } y \rightarrow \infty \quad (25)$$

where u and v are the flow velocity in x- and y-direction, respectively. T is the temperature.

Supplementary Figure 48. Problem domain and boundary conditions.

The initial relative humidity of the environment is assumed to be 20%. A simple approach is to assume that the interfacial temperatures are the same in both phases. The ambient temperature is set at 273.15 K.”

5. Finally, English language as well as several typos must be improved and corrected.

Response: Thanks for the meaningful comments. We apologize for the language issue and the typos of the previous version of manuscript. We have now worked on both language and readability and have also involved native English researchers for language corrections. We really hope that the writing style and language level have been substantially improved.

Reviewer #3 (Remarks to the Author):

The manuscript by Xu et al. on the effects of dehydration on the structural reorganization of dynamic hydrogels covers an interesting and emerging topic in the community. The authors specifically study the mechanical deformation and structural rearrangements observed in boronate ester hydrogels undergoing dehydration. While the topic and many of the observations are interesting, the manuscript is rather dense and obtuse in phrasing and it is difficult to extract

the main conclusions or hypotheses from the document in its current form. As such, it is premature for publication in Nature Communications. However, given the interest in this topic and its relatively unexplored nature, the authors should be encouraged to restructure and reformulate their work to make the following points more clear.

Response: We appreciate the insightful and constructive comments and advice provided by the reviewer. We have carefully considered these concerns and have undertaken a proper revision of the manuscript.

1. What are the central hypotheses and conclusions of the work in simple, accessible language?

Response: Thanks for the meaningful comments. In brief, we present the distinctive anisotropic dehydration modality of dynamic hydrogels, which is fundamentally different from often observed isotropic dehydration of covalent hydrogels. The hypothesis is that the reorganization of dynamic networks in dynamic hydrogel regulates the macrostructure evolution during the dehydration process, resulting in a novel dehydration modality of materials. The main conclusion proposes a criterion based on the competition between cohesion energy and adhesion energy, which determines the dehydration modality of dynamic hydrogels. Such description can also be found in the Abstract and in the manuscript on Page 17:

“Unlike the typical isotropic contraction of covalent hydrogels during dehydration, a fast reorganization of dynamic hydrogels facilitated the migration of polymer chains under the dehydration induced stress. Hollow xerogel tubes with unique wall-knot structures can be generated from dynamic hydrogels, based on a criterion that considers the competition between cohesion energy and adhesion energy. These tubes can be used to engrave nanopatterns onto a substrate through macroscopic structural reconstruction.”

2. What are the broader implications of these hypotheses and conclusions? What excites a general audience as for Nature Communications as opposed to a more focused polymer processing community?

Response: Thanks for attentive advice. Regarding the wider implications of the current research, they can be found throughout the manuscript.

a). The experiments are conducted using a universal dehydration process. The phenomenon of dehydration, which occurs naturally, is particularly important to consider when working with materials that contain water, such as hydrogels, as well as other solvents or evaporating components. Hence, the current research has universal implications and significant practical applications.

b). The main hypothesis is that the reorganization of dynamic networks in dynamic hydrogels regulates the macrostructure evolution during the dehydration process, resulting in a novel

dehydration modality of materials. This hypothesis is based on the intrinsic properties of dynamic hydrogels, which have been further confirmed in different dynamic systems. Therefore, this finding is novel and universal for dynamic hydrogels and materials.

c). The main conclusion proposes a criterion based on the competition between cohesion energy and adhesion energy, which determines the dehydration modality of dynamic hydrogels. This conclusion is independent of dynamic materials or substrates, and is based solely on the competition between them, as determined through our theoretical analysis. Therefore, our findings are promising for application to other systems.

d). Dynamic materials are prevalent in nature and form the basis of living organisms. Structures such as hollow tubes with knots, like bamboos, are particularly unique. The current findings offer a new perspective on understanding complex natural phenomena in terms of physicochemical processes.

Herein, our research work is multidisciplinary and of high-quality, extending beyond the area of polymer processing. Our work provides a detailed and comprehensive description of a universal phenomenon originating from dynamic hydrogels. We present herewith reasonable hypotheses and conclusions that are significant even for a general audience and can inspire researchers from different fields.

Corresponding description regarding the findings, hypothesis/conclusion and their implications can also be found in the manuscript:

“The dehydration process is widely recognized as a significant phenomenon in nature. Hydrogels, which are important functional materials with high water content and crosslinked networks, encounter the issue of dehydration in their practical applications.” (Page 1)

“However, dynamic hydrogels, distinguished by their reversible networks, possess physicochemical properties analogous to those of organisms, and exhibit adaptive responses to stress^{15,16}. Pitifully, the current understanding of hydrogel dehydration is primarily based on static systems. This has rendered it challenging to investigate macroscale transitions in dynamic hydrogels, which are more common in living systems and functional materials.” (Page 2-3)

“This result could be induced by the competition between cohesion energy and adhesion energy. For example, in the virgin dynamic hydrogels, as the monomer concentration increased from 0.5 M to 4 M, the cohesion energy exceeded the adhesion energy in the plastic tubes, so that at 4 M monomer the xerogel formed with isotropic shrinkage. In comparison, when the substrate was replaced with glass tubes to enhance the hydrogel-wall interface interaction, the anisotropic hollow tubes appeared again due to the enhanced adhesion energy at the interface (Supplementary Figure 14).” (Page 9)

“Anisotropic dehydration, which relies on dynamic polymer networks, was found to be widespread among various materials (Figure 4).” (Page 15)

“Unlike the typical isotropic contraction of covalent hydrogels during dehydration, a fast reorganization of dynamic hydrogels facilitated the migration of polymer chains under the dehydration induced stress. Hollow xerogel tubes with unique wall-knot structures can be generated from dynamic hydrogels, based on a criterion that considers the competition between cohesion energy and adhesion energy.” (Page 17)

“The potential applications of large-scale curved surfaces produced through universal anisotropic dehydration with excellent conformality are significant, such as in generating metasurface and for nanotransfer printing.” (Page 18)

3. How do the underlying physics of the system relate to the dynamics of the system? This is alluded to but not studied clearly and in detail and this is the main advantage of using a boronate ester-based system as it is highly tunable in its dynamics.

Response: We thank the reviewer for the suggestive comments. The distinctive anisotropic dehydration modality of dynamic hydrogels is demonstrated to originate from the dynamic nature of the system, as discussed in the current manuscript and shown by using other dynamic crosslinkers. The conclusion is based on a rigorous theoretical analysis and a comprehensive examination of the relevant experimental data.

a). The migration of polymer networks is dependent on the dynamic reorganization of the system. This hypothesis is also confirmed when compared to covalent hydrogels, in which the quiescence of networks led to failure of the hydrogel/inner wall interface under the stress induced by dehydration, and later to isotropic shrinkage.

b). As shown in experimental evidence, the introduction of a fresh-cut surface, which exhibits greater dynamic properties than the original gel, plays a crucial role in tuning knot position. Consequently, the anisotropic dehydration modality observed in dynamic hydrogels is highly reliant on the dynamic properties of the system.

c). In a comparable manner, a dynamic hydrogel based on a Schiff base is employed in the same way. Compared to crosslinked networks with boronate esters, xerogels crosslinked with Schiff base contained failure at the hydrogel-wall interface, which behaved as the shrunken and curved surface around the knot. It could be induced by the longer relaxation time scale of dynamic crosslink sites (Supplementary Figure 24). Therefore, the fast reorganization of dynamic hydrogels during dehydration predominantly contributed to the transition of xerogels from isotropy to anisotropy, while a fast dissociation of the crosslinkers plays a critical role.

Therefore, the observed physicochemical phenomenon is undoubtedly closely related to the dynamic characters of the system.

The principal advantages of boronate esters are also outlined in the Supplementary Information, as detailed below.

Supplementary Figure 1. A dynamic reorganization in dynamic hydrogels based on boronate ester with external force. Boronate esters are characterized by tunable mechanical properties and a thorough investigation of their properties^{1, 2, 3}.

4. Finally, the manuscript text and figures should be clarified to be accessible to a general audience. Here, less is often more, and making sure that the central points are communicated clearly is more powerful than an overwhelming amount of information and at times overly vague text.

Response: Thank you for your helpful comments. We apologize for the confusion caused by the previous version of the manuscript and sincerely hope that the improved readability will make our text/content easier to follow.

Reviewer #4 (Remarks to the Author):

In this work, the authors investigate the dehydration behaviors of covalent or dynamic cross-linked hydrogels filled in hollow tubes. They revealed that the dynamic cross-linked gels dehydrated to give the xerogels unique wall-knot structures by reorganization of the cross-linked network when their adhesion energy with the surface is higher than their cohesion energy. The authors well investigated the validity of their works through rheological measurements, structural

investigations, and simulation. However, there is still ambiguity in the experimental details. Some concerns ought to be given attention, as follows:

Response: We would like to express our gratitude to the reviewer for the constructive feedbacks on this study. We are pleased to learn that the reviewer fully acknowledged the significance and originality of this study. The following point-to-point response is intended to provide comprehensive details regarding the experiments and to avoid any misleading descriptions.

1. I would like to know how to take out the xerogels with wall-knot structures from the hollow tubes. Is it possible for the xerogels with wall-knot structures to detach from the surface of the tube due to the cohesive energy exceeding the adhesive energy by the increase in concentration during the drying process?

Response: Thanks for the valuable comments. Based on our experience in experimental operations, it can be stated with certainty that the xerogels will not spontaneously move out from the tubes after dehydration, regardless of the type of tube used. In the typical procedure, a thin metal rod is employed to gradually dislodge the xerogels from the tubes by pushing the knot. Such description can also be found in the manuscript on Page 20:

“In the typical procedure, a thin metal rod is employed to gradually dislodge the xerogels from the tubes by pushing the knot after the hydrogel has been dehydrated.”

We believe that during dehydration, cohesive energy is more than adhesive energy after a certain time (Supplementary Figure 45). With regard to the detachment between xerogel and the inner surface of the tube, it is possible that this may occur during an increase in cohesive energy. For the typical xerogels derived from dynamic hydrogels, we have examined the inner diameter of the tubes and the outer diameter of the xerogel tubes. The differences observed are below 3% and the absolute values are on the scale of several hundred micrometers. Such differences may be related to the detachment of the xerogel on the inner wall. Nevertheless, given the negligible difference in experimental operations, it is not necessary to provide a more detailed discussion in the main manuscript. Such description can also be found in the manuscript on Page 20:

“The difference between insider diameter of tube and outside diameter of hollow xerogel tube was negligible (Supplementary Figure 32).”

The detailed information is provided in the Supplementary Information section, as described below,

Supplementary Figure 32. The diameters of the tubes and the resulting xerogels. The data were collected using vernier caliper. Due to the deformation induced by measuring force and Abbe error, it can be assumed that the difference between ID and OD is negligible. Scale bar is 5 mm.

2. Some xerogels have narrow parts near the knot while other xerogels do not show the narrow parts (ex. Figure 2f). What factors determine which structures are obtained?

Response: We are grateful for the Reviewer's perceptiveness. Some xerogels with narrow parts near the knot were found in specific systems, like in Figure 2f and Figure 4f. The underlying mechanism of such phenomena may be attributed to the varying relaxation times observed in different systems. The elevated monomer concentration (Figure 2f) or the relatively low dynamic system (Schiff base system, Figure 4f), both with relatively longer relaxation time, impede the release of stress induced by dehydration. In particular, the fluctuation of the monomer fraction around the knot (the late stage of dehydration) can be acute, resulting in a reversal of the competition between cohesive energy and adhesive energy. The detachment between the xerogel wall and the tube then occurred. As a consequence, the shrunken and curved surface around the knot was observed in the system. Another experimental phenomenon also lent support to this theoretical hypothesis. When glass tubes were employed as the substrate, the formation of xerogels with narrow parts around knots was not observed, due to the higher cohesive energy present. Corresponding description can also be found in the manuscript on Page 16:

“Compared to crosslinked networks with boronate esters, xerogels crosslinked with a Schiff base contained failure at the hydrogel-wall interface in the vicinity of the knot, which behaved as the shrunken and curved surface. This could be induced by the sharp fluctuation of the monomer fraction around the knot in the late stage of dehydration and the longer relaxation time scale of

dynamic crosslink sites (Supplementary Figure 24). Therefore, the fast reorganization of dynamic hydrogels during dehydration predominantly contributed to the transition of xerogels from isotropy to anisotropy, while a fast dissociation of the crosslinkers plays a critical role (Figure 2).”

Otherwise, the modality of hydrogel dehydration follows the general description as illustrated in Figure 1 and 2. For dynamic hydrogels, if the adhesive energy exceeds the cohesive energy, hollow xerogel tubes can be generated after dehydration. Conversely, the dynamic hydrogels in tubes are turned into solid xerogels after dehydration. For typical covalent hydrogels, the transformation into solid xerogels occurs as a result of dehydration.

3. The authors should show the details of how to obtain the xerogel surface with the aligned sub-10 nm patterns. Is it possible for similar methods to be applied to the normal hydrogels or xerogels without hollow tubes? That is important to consider whether this method is novel.

Response: Thank you for the suggestive comments. The xerogels with the nanopatterns were generated using the standard procedure in this work. Only, plastic tubes with nanopatterns were used as the substrates for the formation of xerogels. The detailed description can be found in the manuscript on Page 20:

“Generation of xerogels with nanopatterns on the surface

To obtain the xerogels with nanopatterns on the surface, a cylindrical plastic tube (1 mL disposable syringe Injekt® F, Carl Roth, Germany) with an internal diameter of 4.60 mm was used as the model for the molding of the hydrogels. As the commercial products, the inner surface of the tube with a series of trenches was identified (Supplementary Figure 27). Following dehydration, the dynamic hydrogels that had formed in situ within the plastic tubes underwent a transformation, becoming hollow xerogel tubes. The nanopatterns were observed on the outer surface of the xerogel tubes, which corresponded to the inner surface of the plastic tubes.”

Supplementary Figure 27. The surface morphology of the inner surface of plastic tubes with nanopatterns. The SEM image (false color), AFM image and height information of the inner wall surface with patterns are presented.

As for the Reviewer's question regarding the applicability of similar methods to the normal hydrogels or xerogels without hollow tubes, we concur that these points are related to the novelty of our research. Firstly, we have demonstrated the generation of nanopatterns through the use of conformality in hydrogels, which is in line with the intrinsic properties of hydrogels as previously discussed in the manuscript. In essence, this is a contact printing method based on the template. Therefore, our method could be applied to normal hydrogels, including those belonging to the boronate ester category, which are known as dynamic hydrogels.

For xerogels without hollow tubes, or what we referred to as bulk hydrogels, we posit that this concept has minimal bearing on our proposed approach. There are several methods reported on constructing nanopatterns on hydrogel surface, including direct writing (*Advanced Materials* **2013**, 25, 41, 5869-5874; *Small* **2011**, 7, 2, 226-229) and template (*Langmuir* **2006**, 22, 3, 1369-1374; *Chemistry—A European Journal* **2002**, 8, 23, 5363-5367). All these methods are promising in

generating nanopatterns on bulk hydrogel surface. Nevertheless, in comparison with previously reported methodologies, our approach demonstrates a distinct advantage in the generation of patterned, **curved surfaces** on a **large scale** with **high resolution**. The approach demonstrates considerable potential for nanopattern transfer, by using universal materials, low energy consumption, non-planarity and high resolution. Therefore, our methods are highly novel and original, with great application potential in generating metasurface and for nanotransfer printing. In contrast, those reported works demonstrate superior capabilities in generating nanopatterns in bulk hydrogel systems. Such description can also be found in the manuscript on Page 16-17:

“Creating nanopatterns on a large-scale surface is often challenging and usually requires complex devices or materials^{35, 36, 37}, especially for curved surfaces. Taking advantage of the good conformality of the dynamic hydrogels within the molds, fine structures on the inner walls were transferred onto the outer surface of the resulting xerogels after dehydration.”

“The nanostructures created on a 3D curved surface on a large scale by anisotropic dehydration of dynamic hydrogels were demonstrated, e.g., the length of hollow xerogel tubes with nanopatterns could be up to ~10 cm, while the diameter could be up to ~1 cm (Supplementary Figure 29). Therefore, the external surface of the centimeter-scale hollow xerogels can be used to complementarily engrave structural features of the curved substrates. Given this, the anisotropic dehydration of dynamic hydrogels has great potential in various applications for constructing nanopatterns on curved surfaces on a large scale.”

4. The hollow tubes with various diameters were used in this work. What effect does the diameter have on the structure obtained?

Response: We thank the reviewer for the attentive question about the relationship between tube diameter and the resulting xerogels. As we described in the manuscript, the generation of xerogels is based on the migration of polymers. Thus, tube diameters can be used to tune the xerogel wall thickness by regulating the accumulation of polymers. In reverse, it can also confirm our hypothesis that the migration of polymers dominates the formation of hollow xerogel tubes during dehydration process. Corresponding description can be found in the manuscript on Page 16:

“It was also found that the different xerogel wall thicknesses could be achieved in different systems by adjusting the diameter and polymer fraction, because the generation of xerogel wall was contingent upon the accumulation of polymers along the radial direction (Supplementary Figure 26). Therefore, the dehydration-induced stress, which drives polymer migration, proved to be an effective approach even on a large scale, with broad applicability.”

Supplementary Figure 26. The relationship between the xerogel wall thickness and the applied tube. a). Two tubes of different materials with comparable diameters were utilized. The xerogel hollow tubes exhibited a similar thickness. The monomer concentrations were 2 M. b). Two tubes with different diameters were utilized. As xerogel wall generation is contingent upon the accumulation of polymers along the radial direction, it was found that the different xerogel wall thicknesses could be achieved in different systems by adjusting the diameter and polymer fraction. The initial monomer concentrations are labelled.

5. In Supplementary Figure 4, is it possible to compare water contact angles on surfaces with different curvatures, appropriately?

Response: We are grateful for the constructive comments. To date, we are only able to provide a brief qualitative description, such as the terms 'hydrophilic' or 'hydrophobic', rather than any quantitative description or comparison of surfaces with different curvatures. This topic has been comprehensively discussed in the aforementioned references (41. *Chem Phys*, **2015**, 457, 63-69; 42. *J Colloid Interface Sci*, **2012**, 367, 472-477).

The water contact angles are more complex on curved surfaces. As discussed in previous research works, the volume of water, curvatures and materials have been identified as contributing factors. Even when the materials are identical and the curvatures are the only difference, the apparent contact angle can be complex due to the change in contact area, as shown in the figures below:

[REDACTED]

Figure. Angle of liquid–solid curvature of droplet on concave spherical surfaces plotted as a function of droplet volume for selected principle contact angles. (*Chem Phys*, **2015**, 457, 63-69)

[REDACTED]

Figure. Contact angle value calculated with the proposed procedures (intercepts of the linear fittings) on a carbon steel cylinder and a sphere, compared to the values measured on a flat surface. (*J Colloid Interface Sci*, **2012**, 367, 472-477)

Consequently, it is our contention that it is currently not possible to make an appropriate comparison between the apparent static water contact angles on surfaces with different curvatures. On the other hand, given our knowledge of the materials and their hydrophilicity/hydrophobicity, including the water contact angle on flat glass being below 30° and the water contact angle on flat polypropylene (PP, the raw material of plastic syringes) being above 100°, we can conclude that the integrity of our research is sufficiently robust. We also added the details in Supplementary Figure 4 as below:

Supplementary Figure 4. The static water contact angles of various substrates. The NMR tubes are manufactured from glass. The injector wall is constructed from polypropylene. As the diameters of different injectors are different, the values of static water contact angles only reflect the apparent contact angle. N = 5 individual experiments. Error bars are SD.

REVIEWERS' COMMENTS

Reviewer #1 (Remarks to the Author):

In our opinion, the Authors replied in a satisfactory way to all the questions asked: figures have been improved and more details have been added to the original text; comments about the stress state during the dehydration process have been added; information about the computational model have been added in the Supplementary Information.

So, the article has improved in many of its aspects and we recommend publication.

Reviewer #2 (Remarks to the Author):

Reviewer #3 (Remarks to the Author):

The manuscript demonstrating how anisotropic dehydration in dynamic covalent materials can influence material structure and properties has been substantially improved during the revision. While there are still open questions about the relationship between dehydration and material properties in such dynamic materials, the central points are clear and should encourage future work in this direction. I encourage acceptance of the revised manuscript.

Reviewer #4 (Remarks to the Author):

The authors responded appropriately to all reviewer's comments and questions in the revised manuscript and the response letter. Their findings would contribute to the strategy for the large-scale fabrication of curved surfaces with nanopatterns. I support the publication of the revised manuscript in this journal.